# Review article: Towards a context-driven research: a state-of-the-art
# review of resilience research on climate change
Ringo Ossewaarde[1], Tatiana Filatova[2], Yola Georgiadou[3], Andreas Hartmann[4], Gül Özerol[5], Karin
Pfeffer[6], Peter Stegmaier[7], Rene Torenvlied[8], Mascha van der Voort[9], Jord Warmink[10], Bas Borsje[11]
*Correspondence to*: Ringo Ossewaarde (m.r.r.ossewaarde@utwente.nl)

[1] Department of Public Administration, University of Twente, Enschede, Drienerlolaan 5, 7522NB, Netherlands

[2] Dept of Governance and Technology for Sustainability, University of Twente, Enschede, Drienerlolaan 5, 7522NB, Netherlands

[3] Department of Urban and Regional Planning and Geo-Information Management, University of Twente, Enschede, Hengelosestraat 99, 7514AE, Netherlands

[4] Department of Construction Management and Engineering, University of Twente, Enschede, Drienerlolaan 5, 7522NB, Netherlands.

[5] Dept of Governance and Technology for Sustainability, University of Twente, Enschede, Drienerlolaan 5, 7522NB, Netherlands

[6] Department of Urban and Regional Planning and Geo-Information Management, University of Twente, Enschede, Hengelosestraat 99, 7514AE, Netherlands.

[7] Department of Science, Technology and Policy Studies, University of Twente, Enschede, Drienerlolaan 5, 7522NB, Netherlands.

[8] Department of Public Administration, University of Twente, Enschede, Drienerlolaan 5, 7522NB, Netherlands.

[9] Department of Design Production and Management, University of Twente, Enschede, Drienerlolaan 5, 7522NB, Netherlands.

[10] Department of Water Engineering and Management, University of Twente, Enschede, Drienerlolaan 5, 7522NB, Netherlands.

[11] Department of Water Engineering and Management, University of Twente, Enschede, Drienerlolaan 5, 7522NB, Netherlands.

**Abstract**
Since the 1970s, Holling's socio-ecological systems (SES) approach has been the most predominant
theoretical force in resilience research with regard to the climate crisis.  An overview of the scholarship
in the social sciences during the past five decades reveals two different re-appropriations of Holling's
legacy, which can broadly be classified as naturalist and constructivist, respectively. Characteristic for
naturalist resilience research is its indebtedness to the concepts, methods, and assumptions of the so-
called 'life sciences'. This has resulted in the recasting of Holling's SES into complex systems that are
marked by non-linearity and evolutionary changes. Constructivist resilience research, on the other
hand, relies on the concepts, methods and assumptions that are common in the 'human sciences'.
Accordingly, resilience is studied and critically appraised in its historical, social, and political context.
In this paper, recent developments in resilience research in the social sciences are reviewed to the end
of proposing new research questions. The focus is on the different approaches, models and
commitments that underpin these two approaches to resilience in the context of the ecological crisis.
Particular attention is thereby paid to the naturalist emphasis on adaptation and the constructivist
emphasis on transformation.
Keywords: adaptive resilience, climate change, constructivism, naturalism, SES, transformative
resilience, transformational adaptation
**1.  Introduction**
Crawford Stanley Holling's ecological notion of resilience (Holling, 1973) has become part and parcel
of the social sciences, particularly in the field of social studies of climate change. Some social scientists
have recast and integrated it in their theoretical frameworks. Others accept the terminology and
conceptualization of the term while not necessarily endorsing Holling's theoretical framework. The
ecologist's notion of resilience has been presented as interdisciplinary and thus as having the potential
of building a bridge between the social sciences and engineering (Ostrom, 2007; Thorén, 2014). Holling
corrected what he considered to be an unrealistic view of the world and of ecosystems, namely, as
closed or stable. Against the 'equilibrium-centered' view, he emphasized the influence of random
events (natural or human-caused) on ecological systems (Holling, 1973, 15). Holling's socio-ecological
systems (SES) approach appealed to social scientists since it highlighted the interaction between
human societies (political, social, economic, and technological environments) and natural ecosystems.
Consequently, resilience to climate change, for the social scientist, requires the reformation of
established modes of thought (including conceptualizations of 'nature' and 'society'), lifestyles and
consumer habits, production patterns, health issues, law, economy, science, technology, governance
and politics (cf. Douglas & Wildavsky, 1983; Blühdorn, 2013; Fischer, 2017; Dryzek & Pickering, 2019).

Holling's ecological approach has been adopted by the Resilience Alliance, whose flagship

journal, *Ecology and Society* (established in 1995), provides a platform for SES-based resilience
research. In the social sciences, resilience to climate change has become a research topic since the
Tsunami in 2004 and Katrina in 2005 (Pizzo, 2015). Katrina and, even more recently, Covid-19, social
scientists point out, reveal a vulnerability that does not only consist in exposure to natural hazards.
Instead, what has been made clear is that social, cultural, political, and economic conditions largely
determine the resilience to these natural calamities (Tierney, 2015; Lockie, 2016). In the past decade,
resilience to climate change has been addressed primarily as a policy discourse. Indeed, since 2010, in
the wake of the global financial crisis (2007-2008), global governance actors and national and local
governments – including the Rockefeller Foundation – have had profuse recourse to the language of
resilience. The economic and political interest behind such discourses has gained the critical attention
of social scientists (Hilhorst 2018). This has given rise to new resilience research, new outlets (such as
the interdisciplinary journal *Resilience* (established in 2013)), and the establishment of resilience
research programs in universities around the world. This relatively recent development has meant the
diversification of existing resilience research in the social sciences. As a result, many publications of
the past decade address different definitions and understandings of resilience. Such diversity
corresponds to the diversification of approaches in the social sciences. Resilience research in the social
sciences had been predominantly naturalist. Today, social scientists are increasingly addressing climate
change and resilience to climate change from constructivist angles.
The aim of this paper is to provide an overview of the current state of resilience research with
regard to climate change in the social sciences and propose a research agenda. Current research can
broadly be classified into two main schools of thought, namely, naturalist and constructivist. The latter
is a more recent development in resilience research where the natural sciences and mathematics have
tended to be authoritative. The diversification of resilience research in the social sciences is thus
addressed in the first section of this paper. Naturalism and constructivism are presented as two (social)
scientific approaches underpinned by different epistemological and ontological commitments. It is
suggested that social scientific inquiry into resilience in the context of climate change could be raised
to a next level if these two different approaches meet and interact. To this end, we reconstruct
contemporary debates in that particular field of studies and distil recurrent research topics that divide
social scientists. The issues of adaption and transformation in the context of severe disturbances or
shocks that come with climate change (such as hurricanes, floods, drought, and heatwaves) appear to
be such divisive topics. Finally, naturalist and constructivist directions, as well as possible cross-
fertilizations of these two currents, for future resilience research are identified. We point out that
future resilience research in the social sciences – that is, the types of questions raised, theoretical
frameworks and modes of analysis – will also be determined by changing conditions (ecological,
political, and socioeconomic).

**2. The diversification of resilience research**
One of the earliest appearances of the term resilience – in European literature at least – seems to have
been in one of Aesop's fables, namely, that of *The Oak Tree and the Reeds*. According to one of the
versions of that story, the Oak Tree becomes uprooted during a storm while its fellow reeds survive it.
In a conversation, the Oak Tree expresses its bewilderment that the fragile reeds were able to resist
such a mighty storm while it succumbed. The reeds reply that it is precisely their non-resistance that
saved them. Through their capacity to bend, they moved with the direction of the wind (which thus
did not break them) and rose again when the storm was gone. They were flexible enough. The reeds
'bounce' back and are thus 'resilient'. Indeed, the English word resilience derives from Latin (*resilire*),
which generally meant rebounding. This Latin word can be found in the writings of Seneca the Elder,
Pliny the Elder, Ovid, Cicero, and Livy;  to rebound is also the sense in which *resilire* is used by Cicero
in his *Orations* (Alexander, 2013). The term also appears in Lucretius' *On the Nature of Things*, where
it denotes 'being forced back by a resisting surface […] with reference to the action on Nature' (Pizzo,
2015). Along this line, nature compels all things to 'spring off'.

Despite the various meanings attributed to the term, the connotation attached to *resilire* was

commonly that of rebounding. Up to the early nineteenth century, this was the predominant
understanding of resilience in common language and imagination. A slight shift appeared when
engineers started to use the term to refer to the properties and capacities of materials to absorb
tensions and release energy, and recover their original forms, without breaking or disfiguration after
undergoing some external shock or disturbance (such as  extreme weather conditions) (Estêvão,
Calado & Capucha, 2017; Bergström, 2018; Davoudi, 2018). In the 1950s, psychologists re-adapted the
common sense of the term to mental health and used it to study the coping mechanisms of
concentration camp survivors. Later, the concept is used to study all sorts of trauma, misfortune,
adversity, stress, and mental recovery (Bourbeau, 2015; Estêvão, Calado & Capucha, 2017; Bergström,
2018; Schwartz, 2018). In the 1970s, the ecologist C.S. Holling (1973: 14) redefines resilience as 'a
measure of the persistence of systems and their ability to absorb change and disturbance.' Thus
understood, resilience is widely conceived as the opposite of vulnerability, which is defined as the
inability to absorb change and disturbance (Gallopin, 2006; Miller et al, 2010) - for instance, a coastal
system that is vulnerable to accelerated sea-level rise is not resilient enough (Smit, Goosen &
Hulsbergen, 1998). In such discourses, greater resilience means becoming less vulnerable to change
and shocks. That said, a system can still be vulnerable to other changes while being resilient in other
respects (Gallopin, 2006). Holling incorporates resilience in a socio-ecological systems (SES) approach
to analyze the stability and strength of ecological systems, which are constituted by the interaction
between natural ecosystems and human societies (Alexander, 2013; Bergström, 2018; Béné et al,
2018; Hoekstra, Bredenhoff-Bijlsma & Krol, 2018). Ecosystems, as noted earlier, are rarely closed
systems, but are instead subjected to natural and human influences.

In the social sciences, resilience research that has emerged from Holling's SES approach has

developed along two different lines, which can be called naturalist and constructivist, respectively
(Miller et al, 2010).  These two currents of research have different focuses, raise different questions,
and have recourse to different methods. The naturalist line of research is indebted to the accepted
methods and assumptions of the natural sciences. It has a predilection for mathematical and
simulation models. Social scientists dealing with resilience to climate change research questions
consider resilience as a property of a system, which can be (made) weak or strong. Society is modelled
as a social system that consists of parts (including agents and technologies) and physical properties
that can be objectively studied (Aiken, 2006; Floridi, 2017). Resilience as a system property is an
objective measure of the dynamic equilibrium, stability, strength, or survivability of a socio-ecological
system, including coastal systems, urban systems, forest systems, etc. (Hoekstra, Bredenhoff-Bijlsma
& Krol, 2018).

The naturalist approach to problems that arise through climate change can be very useful,

especially when both the problem and the solution are quite uncomplicated (and hence are primarily
of a technical nature, such as water purification, for instance). The story becomes more complicated
when, for instance, attempts to make communities more resilient to climate change overlook the
political and cultural reasons why particular groups are more vulnerable to the effects of climate
change. Since a model cannot include these reasons, the naturalist social scientist necessarily leaves
out factors that are part of the problem and the solution. In so doing, naturalist social scientists may
well become unwitting allies of political powers and help to perpetuate status quos. Constructivist
social scientists have shown increased interest for resilience research precisely because resilience is a
term profusely used by global and national powers during the last two decades.

Historically, constructivism in the social sciences has arisen in reaction to what was experienced as

the narrowness of the naturalist approach. The constructivist does not believe that reality is so
objective that it can be fully grasped and (s)he does not try to objectify it. Instead, natural and social
phenomena can only understood by taking into account diverse human perceptions, experiences,
meanings, interests, values, identities, patterns of domination, etc. Constructivist social scientists thus
think that it is mistake to compress the social sciences into the mold of the natural sciences. In
resilience research, they typically model society as a historically embedded construct that is the result
of particular understandings of nature, society and the person, of values, symbols and historical
practices (which may not be very rational or just), and power relations. Constructivists tend to be more
critical and politically sensitive. They are generally more aware of the potential and actual abuse of
power. When addressing resilience issues in the context of climate change, they typically express
concern for vulnerable communities. Research topics thus include the(un)equal distribution of
environmental burdens, struggles for recognition, claims to participation, and unequal impacts of
anthropogenic climate change (Braun, 2014; Yanarella & Levine, 2014; Skillington, 2015; Sjöstedt,
2015; Weichselgartner & Kelman, 2015; Pizzo, 2015; Lockie, 2016; Derickson, 2016; Lyster, 2017;
Schlosberg, Collins & Niemeyer, 2017; Mummery & Mummery, 2019). Davoudi (2018: 5), for instance,
introduces the notion of 'unjust resilience'. Unjust resilience refers to absorption of changes or
disturbance through a systematic neglect of vulnerable groups and marginalized people. Katrina and
the Covid-19 crisis reveal such systematic injustice. And Glaser et al (2018: 3) refer to 'undesirable
resilience', 'bad resilience' and 'wicked resilience'. These are notions that emphasize how resilience
may go hand in hand with the enforcement of an undesirable or unjust condition. The resilience of
oppressive systems (like tyrannical regimes) that systematically marginalize, discriminate or persecute
certain groups are an example of this.

**2.1. The naturalist view on resilience**

In the social sciences, naturalist research as such arose in the context of the development of
cybernetics, computational power and automation (and automated decision making) (Simbirski, 2006;
Floridi, 2017; 2018; Davoudi, 2018). Naturalist social studies are based on the cybernetic idea that
machines, organisms, and societies show considerable similarity in structure and function; and can be
described in terms of systems. Since the 1940s, such studies have typically adopted cybernetic
complexity theory as their distinctive overarching theoretical outlook, within which other theories (for
instance, on behavior change, on decision making under risk, or on social institutions) are
incorporated. In complexity theory, machines, organisms, and societies are modelled as complex, non-
linear, evolutionary systems. Complex systems are composed of many components, including
properties, agents, resources, and governance systems. All components interact with each other, in
response to ever-changing environments and disturbance (Walsh-Dilley & Wolford, 2015; Juncos,
2017; 2018). From this naturalist point of view, resilience to climate change is a matter of evolution:
resilience is 'evolutionary resilience' (Pizzo, 2015: 137; Davoudi, 2018: 4).In the 1970s, naturalist social
scientists incorporated Holling's notion of resilience within their own cybernetic complexity theory and
cybernetic methodology (Wiese, 2016; Bergström, 2018). That is, socio-ecological systems are
cybernetically conceptualized as adaptive complex systems. The ability to cope with uncertainty and
complexity is one of the capacities of individual agents and interacting agents. The latter are able to
interact and self-organize, learn and adapt (in an incremental or transformative way), making the
system flexible in absorbing shocks and developing in face of changes (Jesse, Heinrichs &
Kuchshinrichs, 2019).
Naturalist social scientists tend to emphasize a type of laissez-faireism, pointing out that
adaptive complex systems have their own self-organizational structures that should not be interfered
with. Bureaucratic interventions to address vulnerability and increase resilience to climate change
typically generate unintended consequences that may well reduce a system's ability to absorb changes
and disturbances (Adger et al, 2011). In 2001, Holling introduced the notion of 'panarchy' as an
alternative to hierarchy, to safeguard the self-organization of complex systems against the threat of
bureaucratic intervention (Holling, 2001). Derived from the ancient Greek god of the woods, Pan,
panarchy refers to the structure in which complex (ecological and social) systems are interlinked in an
evolutionary process of adaptive cycles of growth, accumulation, restructuring, and renewal (Berkes
& Ross, 2016). Accordingly, when confronted with shocks (like extreme weather events), adaptive
systems stabilize with supporting self-organizing structures until those structures are overstretched
and can no longer absorb changes and disturbances; this is when there is a transformation of the
system(Allen et al, 2014). In other words, in naturalist research, the notion of panarchy (as an
evolutionary mode of system self-organization) complements Holling's earlier notions of socio-
ecological systems and resilience (as a system property). In Holling's naturalist theory of panarchy,
resilience is a primary system property that is measured by the magnitude of shocks that can be
absorbed before the structures of system change  (Boyer, 2020).
Methodologically, naturalist social scientists have typically embraced agent-based modelling
(ABM) as their favorite mode of analysis in resilience research. They focus on the constant refinement
of simulation tools (that can cope with complexity, uncertainty and multiplicity of agents) and
techniques of regulation in favor of adaptation (cf. Cote & Nightingale, 2012; Patriarca et al, 2018).
Since the 1970s, when it emerged from mathematical sociology, ABM has been a much endorsed tool
used in complexity-theoretic research for analyzing complex systems. (Conte & Paolucci, 2014). ABM
is a computational mode of analysis that simulates complex (non-linear) systems that include diverse
interacting agents that make decisions, interact and learn or adapt in their ever-changing environment,
according to programmable rules . (Hawes & Reed, 2006; Farmer & Foley, 2009; Van Duinen et al,
2015; Martin & Schlüter, 2015; Sun, Stojadinovic & Sansavini, 2019). ABM computes, in probabilistic
terms, the recovery process of complex (non-linear) systems under stress and tracks the emergence of
new stages, phases or entries into new adaptive cycles (Filatova, Polhill & Van Ewijk, 2016). In the social
sciences, naturalist scholars calculate resilience to climate change at the system level as a system
property (Pumpuni-Lenss, Blackburn & Garstenauer, 2017). Since ABM traces feedbacks between
micro-macro scale explicitly, ABM also enables naturalist scholars to  estimate the resilience of a
system's individual agents, communities or (sub)groups of agents.

**2.2 The constructivist view on resilience**


In the social sciences, constructivist resilience research is also inspired by Holling's SES

approach. Yet, for constructivists, resilience to climate change is not a system property. It is instead  a
socio-political construct that is created by diverse stakeholders (Walsh-Dilley & Wolford, 2015;
Weichselgartner & Kelman, 2015; Kythreotis & Bristow, 2017). Constructivist research includes a
variety of (typically phenomenological and discursive) scientific perspectives. Constructivist resilience
research primarily focuses on the political context and socio-political implications of resilience
discourses. As a construct, resilience to climate change is not so much technical as political and
administrative in nature (Alexander, 2013; Bourbeau, 2015; Boas & Rothe, 2016; Juncos, 2018; Wessel,
2019). And given its political and administrative nature, resilience is invested with ideology and myth.
Constructivist scholars typically stress that resilience is a neoliberal construct. That neoliberal ideology
manifests itself in the belief in adaptive cycles governed by invisible laws and the non-interventionist
stance. It is thereby overlooked that the so-called self-organizing system is itself the result of political
decisions over a long period of time. Constructivists thus point out that resilience has become a
buzzword for governments that seek to shift the responsibility for vulnerable systems, floods,
pollution, safety, welfare, health, etc. to 'resilient' individuals. Governments, in these cases, have
recourse to resilience to make individuals more self-reliant (or less dependent on the government)
when it comes to coping with their own struggles in dealing with the challenges of climate change
(Braun, 2014; Pizzo, 2015; Tierney, 2015; Howell, 2015; Anderson, 2015; Ksenia et al, 2016; Schwartz,
2018; Davoudi, 2018). For instance, governments that fail to provide basic access to water to millions
of rural citizens advocate community-based water management schemes, the leading paradigm for
rural water access in East Africa (Katomera & Georgiadou, 2018). Such schemes 'work' for the state
(and donors) as a means of shifting (or offloading) responsibility for public service provision to the
most vulnerable citizens for whom community management may not be a preferred option (Katomero
& Georgiadou, 2018).

Constructivist scholars tend to critically analyze resilience as an ideological construct. Such

critical studies are typically inspired by the works of Michel Foucault, in the sense that resilience is
analyzed as a discursive construct or ideological discourse. For Foucault, a discourse refers to systems
of thoughts and beliefs, expressed through language and practices that systematically construct
subjects and societies of which they speak. In other words, both language and practices are creative
acts. Language is not a neutral tool of communication. Through resilience discourses, a particular type
of subject (like resilient or self-reliant rather than vulnerable or dependent citizens) and a particular
type of society (like a market-based 'society') are discursively constructed and reinforced (Miller et al,
2010). Evans and Reid (2013) argue that as a discursive construct created by power holders, resilience
has the character of a doctrine, according to which the resilient subject must constantly adapt to a
dangerous and changing world, and is willing to accept this. Given this doctrine, vulnerability is rejected
as weakness, a moral flaw (like a lack of character or a lack of will power) or simply illegitimate (the
ability to absorb shocks being the new norm). Many critical constructivist scholars see the political
reactions to events like Katrina (2005), Fukushima (2011), and Covid-19 (2020) as manifestations of
such ideology. A problematic normativity is brought into existence when citizens are told that they
must adapt to ecological and societal catastrophes, and when vulnerable citizens are left abandoned
by their government as they are expected to be self-reliant (Fainstein, 2014; Tierney, 2015; Ribault,
2019). Constructivist scientists also stress that such catastrophes present themselves as
'anthropological shocks' (Beck (2015: 80). Such shocks may open up counter-discourses that contest
domination (Fazey et al, 2018). Katrina, for instance, proved to be such an anthropological shock
because it opened up a counter-discourse that brought up the issues of colonial patterns of racism,
slavery, vulnerability, and abandonment (Beck, 2015). As an anthropological shock , it is a potential
initiator of policy transformations beyond the resilience discourse.
Constructivist scholars not only emphasize the role of neoliberal ideology that legitimizes
established power relationships and patterns of domination in resilience discourses. They also point at
the role of myth and myth-making in the discursive construction of resilience. Constructed as a myth,
resilience is understood as a widely embraced narrative. Resilience is a story that connects diverging
ideologies, values, interests, worldviews and power relations. The 'myth of resilience' (Kuhlicke, 2013)
refers to the stories that stakeholders enact to make sense of the radically surprising discovery of
something entirely unknown (like Katrina or the Covid-19 crisis). As narrators, stakeholders interpret
their own capacities to deal with stresses and shocks, such as extreme weather events (like floods,
droughts, and heatwaves). In this context of making sense of an unknown phenomenon, stakeholders
develop the capacity to adapt and transform through mythmaking. For instance, the increasing
attention on 'urban climate resilience' (Tyler and Moensch, 2012) resonates with the myth that cities,
or 'local governments', are to lead and shape climate change adaptation as a form of bottom-up self-
organization for absorbing changes and disturbances (O'Hare et al., 2016; Klein et al., 2017).


**3. Bridging the naturalist and constructivist view on resilience**

In the social sciences, naturalist and constructivist resilience research are based on contrasted
premises, each having their own theoretical and methodological outlooks. Given such scientific
contrasts, it has been widely questioned whether resilience can possibly operate as a theoretical model
or unifying paradigm – and whether such a unifying paradigm would be desirable in the first place
(Alexander, 2013; Thorén, 2014; Bourbeau, 2015; Fainstein, 2015; Pizzo, 2015). A unifying paradigm is
neither possible nor desirable. Yet, naturalist and constructivist research can be brought together to
enrich and renew understandings of resilience to climate change. Naturalist resilience research has the
great merit that it may help to increase complex systems' robustness to system failure when faced
with shocks and disturbances. ABM – a mode of analysis that complexity theorists tend to prefer – may
be a valuable tool for developing procedural stability, environmental risk management under
conditions of uncertainty, provision of planning security, and prevention of adverse consequences
from disruptive shocks (Schilling, Wyss & Binder, 2018). Constructivist resilience research provides a
critical and most penetrating understanding of resilience as a construct (first of all, a discursive
construct, myth or narrative) that contains political intention and direction. Its interpretation of
resilience to climate change is useful for generating understanding of how resilience is mobilized, taken
up in climate governance, and resisted by social movements' counter-discourses, such as the Fridays
for Future, Black Lives Matter and Extinction Rebellion, that push for less unsustainable trajectories
and for more protection of vulnerable citizens and communities.


**3.1 The debate on adaptive and transformative resilience**

In recent years, the contrast between naturalism and constructivism in resilience research has come
to revolve around the issue of adaptation and transformation (Chandler, 2014; Redman, 2014;
Fainstein, 2014; Dahlberg et al, 2015; Sjöstedt, 2015; Boas & Rothe, 2016; Duit, 2016; Ziervogel, Cowen
& Ziniades, 2016; Clément & Rivera, 2017; Lyster, 2017; Schlosberg, Collins & Niemeyer, 2017; Fazey
et al, 2018; Glaser et al, 2018; Hoekstra, Bredenhoff-Bijlsma & Krol, 2018; Jesse, Heinrichs &
Kuchshinrichs, 2019; Dryzek & Pickering, 2019).  It is an urgent issue that emerges from an ambiguity
in Holling's SES approach (Redman, 2014). In the 1970s, Holling (1973) reinterpreted resilience as
bouncing back or forward in terms of SES adaptation. Adaptation refers, on the one hand, to the
capacity of agents to influence the system (and influence or strengthen resilience as a system
property). And on the other hand, it alludes to panarchical adaptation to new (ecological and social)
environments, as an evolutionary process towards a new stage, phase, or adaptation cycle (Boyd et al,
2015). Yet, as Holling emphasizes, the bouncing back and bouncing forward of a system not only refers
to a return to some previous (dynamic) equilibrium or to the persistence and endurance of systems. It
also refers to socio-ecological transformation in an ongoing process of non-equilibrium and instability
and reinvention of systems in changing environments marked by different adaptive cycles (growth,
accumulation, restructuring, and renewal) (Folke, 2006). Transformation refers to the capacity of
agents to create a new system and a new discourse, particularly when conditions make the existing
system untenable or illegitimate. Constructivist resilience research is primarily focused on
transformation. Such research unsettles taken-for-granted assumptions and definitions of the
situation expressed in established discourses; and it ignites new imaginations and counter-discourses
needed for realizing less unsustainable futures (Fazey et al, 2018). Recently, a middle ground between
adaptation and transformation has been developed, in the form of 'transformational adaptation'
(Pelling, O'Brien & Matyas, 2015; Mummery & Mummery, 2019: 920). Transformational adaptations,
such as green growth or the greening of the established economy refer to changes that are aligned to
the scale of projected, possible and desirable changes within systems that are informed by (ultimately
constructivist) considerations of environmental and climate justice.

The naturalist emphasis on resilience as system adaptation to climate change means that

resilience research focusses on the degree to which complex systems can build capacity for learning,
as a way to respond to shocks or disturbances, embrace evolutionary change, and live with complexity
and uncertainty (Thorén, 2014; Juncos, 2017; Warmink et al, 2017; Béné et al, 2018). Given
unpredictability and uncontrollability, adaptive resilience comes with short-term planning, uncertainty
reductions, incremental and path-dependent changes (Borsje et al, 2011; Haasnoot et al, 2013).
Adaptive resilience – the system's re-stabilizer – is taken as inherently positive, while disturbances and
shocks (de-stabilizers) are taken as negative (Duit, 2016; Lockie, 2016). It is on the basis of the premise
that adaptive resilience is good that naturalist resilience research ties up with climate risk
management, as a way of managing ecosystem services (critical for survival), under conditions of
ecological and societal shocks and disturbances (Boyd et al, 2015; Berbés-Blázquez et al, 2017). For
instance, when confronted with the near flood events of 1993 and 1995 along the river Rhine in the
Netherlands, the Dutch government responded by increasing the flood conveyance capacity of the
large rivers, thereby decreasing flood water levels (Hamers et al, 2015). Since its completion in 2015,
the Room for the River project is considered effective thus far, particularly as its secondary objective
to increase ecosystem values in the river appears successful. Warmink et al (2017) point out that in
Dutch river management, such adaptation responses are typically conservative and within safety
margins. This leads to over-dimensioning and high costs of water engineering works (like flood
defenses).

The constructivist emphasis on resilience to climate change as system transformation refers to

the emergent transformation of systems into something new beyond the status quo (Ziervogel, Cowen
& Ziniades, 2016; Rothe, 2017; Béné et al, 2018). Transformative resilience is defined as the system's
internal capacities, capabilities and relations that enables it to create a new condition marked by a new
discourse (and accordingly, new or different power relationships). Flood protection, for instance, is
typically a governmental responsibility, but, with a new myth, stakeholders can transform an
established situation and realize alternative scenario's in which responsibilities may be distributed
among different stakeholders (Warmink et al., 2017). Adaptive resilience comes with evolutionary
change (the definition of change that naturalist research typically endorses). By contrast,
transformative resilience comes with 'metamorphosis'. This type of change refers to a transformation
of systems that is triggered by anthropological shocks that open up new horizons, reassessments
(including of past ideas, beliefs and practices) and rediscoveries (Beck, 2015; Fazey et al, 2018). The
middle ground of transformational adaptation bridges evolutionary change and metamorphosis, in the
sense that such adaptation attends to broader socio-political processes of transformations (Kates,
Travis & Wilbanks, 2012; Ziervogel, Cowen & Ziniades, 2016). The notion of transformational
adaptation picks up on and challenges the transformative logic of system transformation with
simultaneous system adaptation, based on uncertainty regarding how fast and how far disruptions will
go – or whether sustainable transformations will thrive as political projects at all.

Constructivist social scientists criticize the notion of adaptive resilience for not sufficiently

addressing issues of environmental and climate justice. To address issues of power abuse and
domination, the constructivist argument goes, system reconfiguration is needed: injustice inheres in
the established systems. Naturalist resilience research, however, does not exclude considerations of
justice from scientific analysis. Yet, it identifies justice, like resilience, as a system property. Thus,
enhancing adaptive resilience to climate change may entail liberal principles of equity, fairness and
access to resources and services, so as not to privilege or marginalize certain stakeholders (Redman,
2014; Thorén, 2014; Ksenia et al, 2016; Schlosberg, Collins & Niemeyer, 2017; Bergström, 2018). Yet,
naturalist enquiry into adaptive resilience tens to leave the status quo of systems, including the
problematic Global North-Global South relationship (marked by massive power inequality),
unquestioned (Swyngedouw, 2011; Pizzo, 2015; Clément & Rivera, 2017; Davoudi, 2018; Glaser et al,
2018; Dryzek & Pickering, 2019). In constructivist resilience research, by contrast, the justice question
is placed in a context of broader socio-political processes of system transformation: adaptive systems
can be unjust and oppressive (Fainstein, 2014; Weichselgartner and Kelman, 2015; Huang, Boranbay-
Akan and Huang, 2016; McGreavy, 2016; Ziervogel, Cowen & Ziniades, 2016; Ribault, 2019). Adaptive
responses to shocks and disturbances may blur long term sustainability visions, while dominant (or
dominating) stakeholders typically reify existing climate policy efforts in their (standardized) adaptive
responses (Lockie, 2016; Derickson, 2016; Rothe, 2017; Estêvão, Calado and Capucha, 2017; Ribault,
2019). Kythreotis & Bristow (2017) call this phenomenon the 'resilience trap' – the reinforcement of
established power relations (legitimized by dominant ideologies such as neoliberalism) and
contemporary resilience discourses (Blühdorn, 2013; Redman, 2014; Yanarella & Levine, 2014; Lockie,
2016; VanderPlaat, 2016; Schilling, Wyss & Binder, 2018; Glaser et al, 2018; Ribault, 2019). Hence,
constructivist scholars tend to reject Holling's panarchy concept, emphasizing that transformation
towards more sustainable worlds is not an evolutionary process of adaptive cycles but a political-
administrative phenomenon. The middle ground of transformational adaptation, accordingly, must
include a process of filtering out resilience traps that come with adaptive resilience. Transformational
adaptation includes an understanding that adaptive resilience may well enforce a governance of
unsustainability (cf. Van de Ven, 2017).


**3.2 Transformative resilience and sustainability**

For constructivist scholars, transformative resilience is a post-neoliberal construct that is intertwined
with the notion of sustainability. For constructivist scholars, sustainability is based on the idea that
existing systems can be transformed – with respect to social, cultural, political, administrative,
economic, technological and environmental factors –, with the right governance interventions and
reconfigurations of the ecological and social underpinnings of SES (Pizzo, 2015; Weichselgartner &
Kelman, 2015; VanderPlaat, 2016; Ziervogel, Cowen & Ziniades, 2016; Hughes, 2017; Jesse, Heinrichs
& Kuchshinrichs, 2019). Currently, the sustainable energy transformation is no doubt the best example
of such a reconfiguration (Park et al, 2012; De Haan & Rotmans, 2018). Fossil energy sources like coal,
oil and gas are largely responsible for carbon dioxide emissions, which generate global warming. The
sustainable energy transformation, accordingly, is, amongst other things, a response to climate change
that is potentially transformative in negating and transcending established (climate unfriendly) energy
systems. From the (typically naturalist) perspective of strengthening adaptive 'energy resilience' (Béné
et al, 2018: 120; Jesse, Heinrichs & Kuchshinrichs, 2019: 21) – energy systems must adapt to changing
environments in which high levels of greenhouse gas emissions comes from burning fossil fuels for
electricity, heat and transportation. Energy resilience means that established energy systems can limit
the risk of power outage and continue providing reliable energy supplies at stable costs, even in a
turbulent ecological and political environment (Wiese, 2016). The notion of energy resilience, as a form
of adaptive resilience to climate change, implies that the energy transition, including the use of
renewables, can only go via incremental changes and greening of the established economy, to avoid
system collapse (Berbés-Blázquez et al, 2017; Schilling, Wyss & Binder, 2018). The middle ground of
transformational adaptation includes this adaptationist notion of energy resilience but aligns it to the
scale of desirable ecological and societal changes that are informed by justice considerations and
political direction towards less unsustainable futures. Given that established energy systems
insufficiently respond to ecological and societal challenges of climate change, transformational
adaptation may imply the metamorphosis of energy systems.
From the (typically constructivist) perspective of strengthening transformative resilience,
energy resilience comes with the enactment of the energy system's status quo. This is a status quo
that includes powerful agents that have a vested interest in promoting fossil energy. Such agents use
all sorts of tactics (including sponsoring the climate change denial movement) to secure their
established power position (Stegemann & Ossewaarde, 2018; Szablowski & Campbell, 2019). It enacts
a condition of 'energy injustice', particularly in the Global South. The notion of energy injustice refers
to current energy systems that distribute the ecological and economic benefits and burdens of
established energy systems in unfair ways; dominate, degrade and devalue certain stakeholders; and
exclude certain agents from processes that govern the benefits, burdens and recognitions (Jenkins et
al, 2016; Heffron & McCauley, 2017). The transformative resilience of energy systems, which is tied
up with the notion of 'energy justice', refers to the resistance to and negation of a fossil-based energy
system and its oligarchical power structure (increasing the vulnerability of such a climate-unfriendly
energy system); and the creation of a renewable-based system, energy commons and collaboratives
beyond the energy establishment (VanderPlaat, 2016; Bourbeau & Ryan, 2018; Juncos, 2018; Schwartz,
2018; Acosta et al, 2018; Jesse, Heinrichs & Kuchshinrichs, 2019). The middle ground of
transformational adaptation includes the long-term vision of energy governance (for instance, towards
2050), but it searches for realizing such transformation through adaptations by the status quo.
Transformational adaptation means that the sustainable energy transformation comes with the
change of the energy establishment into agents of sustainability – a change that comes from within
the power complex, for instance, via stakeholder participation (like shareholder activism).


**3.3 AI for resilience and sustainability**

Adaptive resilience to climate change comes with short-term systematic adjustments to a
changing technological environment that is currently increasingly dominated by smart urbanism and
artificial intelligence (AI) technologies. Governance actors like the UN, EU and national governments
have all drafted their AI strategies for the making of an 'AI Revolution'. Such actors present AI as a
leading technology that contributes to resolving resilience and sustainability challenges (cf. Taddeo &
Floridi, 2018). Particularly in naturalist resilience research, AI is identified as a new systems property
that permeates systems to generate productivity gains, improve efficiency, lower costs, predict climate
change stress, track carbon emissions, monitor flood risks, etc. (Rajan & Saffiotti, 2017; Khakurel et al,
2018; Vahedifard, et al, 2019; Miller, 2019; Saravi et al, 2019). Strengthening adaptive resilience to
climate change through AI primarily means that an integrated data system for circulating information
(near) real time among agents needs to be developed. In an AI technological environment, resilience
implies close collaboration between agents (tool/model developers, data stakeholders, community-
level stakeholders, state-level institutions, etc.) (Vahedifard, et al, 2019). AI comes in both for
combining datasets into usable information, as a monitoring method (like change detection
algorithms) as well as a tool for forecasting (for instance likely occurrence of a natural hazard due to
extreme events). Identifying, harnessing, synthesizing, and communicating pertinent yet structured
and unstructured data (weather data, cell phone GPS data, social media feeds, traffic cameras, smart
city sensors, images, videos, audio data, etc.) enables agents to better forecast, prepare for, respond
to, and recover from disturbances and shocks (Rajan & Saffiotti, 2017; Vahedifard et al, 2019). In urban
systems, so-called 'city dashboards' rely on big data and AI when it comes to ordering and visualizing
data through interactive maps and graphs (Kitchen, 2018). By being able to predict (estimate or
forecast) more accurately and learn from past disturbances and shocks, lessons can be learned and
applied in building adaptive resilience against disturbances (Saravi et al, 2019). AI, as for instance used
in city dashboards, quantifies the probabilities of occurrence of extreme events, essential in predicting
and preparing for future natural hazards, such as floods or landslides. For instance, with advances in
machine learning, water availability, ice surfaces and melting rates, saturated soils, pollution,
deforestation, etc. can be more precisely or smartly monitored in space and time so that changes over
time can be tracked. Yet, with monitoring also learning of agents and organizations is needed.
In the social sciences, constructivist scientists tend to have a critical view of AI. They do
recognize that AI may help building transformative resilience, given AI's capacity for anticipating future
events. AI may also play a positive role in phasing out of unsustainable yet adaptive systems.
Governance actors, such as the UN in its AI for good program (2017-), the EU in its AI strategy (2018),
and various national governments in their AI programs emphasize the transformative potentials of AI.
Yet, strengthened adaptive resilience can also weaken the transformative resilience that is needed for
materializing sustainable transformations (Khakurel et al, 2018). From a critical constructivist angle, to
make AI serve transformative resilience requires that the domination of giant AI firms (like Google,
Amazon, Microsoft, Facebook, Alibaba, Tencent, etc.) is kept in check. It requires high levels of
transparency and stakeholder involvement in how algorithms are designed, built and applied. In
constructivist researches, it is frequently argued that although big data can be openly accessible (like
satellite imagery for geospatial and data scientists), big data and AI are often in the hands of giant tech
oligarchs (Miller, 2019; Ossewaarde, 2019) that have a vested interest in the further acceleration and
consumption of technological devices (Khakurel et al, 2018). Because of such an oligarchical power
structure, AI  tends to obstruct transformative resilience, exerting power beyond rule of law and
democratic will and understanding. Such power abuse is found in the many recent privacy rights
violations and scandals (like the Facebook-Cambridge Analytica data scandal (2018) and the many
Google scandals) (cf. Taddeo & Floridi, 2018).
**4. Six upcoming themes in diversified resilience research**

In the social sciences, the bridging of naturalist and constructivist scientific approaches in theorizing
change as system adaptation, transformation, or transformational adaptation triggers new research
themes for the study of resilience to climate change. Theorizing change within and of systems has
become the key issue in resilience research, in the wake of changing societal, ecological, and
technological environments. In naturalist research, resilience to climate change is presented as
'evolutionary resilience' and as 'adaptive resilience'. From this angle, the key issue of changing
environments is the survivability of established complex systems under stress. Change is, accordingly,
evolutionary change. In constructivist research, resilience to climate change is presented as discursive,
ideological, mythical (the 'myth of resilience') and as transformative resilience. The key issue of change
is the overcoming of 'resilience to change', 'resilience traps' and 'unjust resilience' or 'bad resilience'
that the status quo that organize established systems produce. Such overcoming of the establishment
is presented as an indispensable condition for enhancing change. Such change refers to
metamorphosis of systems and comes with transformative politics and climate governance. The
reconciliation of naturalism and constructivism in terms of change can be found in the middle ground
of transformational adaptation, which ties incrementalism to long term sustainability visions. It is a
notion that comes with the search for the conditions and tempo of transformations in different
ecological and societal contexts and adaptative cycles. Ultimately, the overarching challenge for future
research is to ensure that resilience to climate change does not compromise sustainability and
considerations of justice (including, environmental, climate and energy justice).
A first promising direction for future resilience research concerns the reconciliation of
naturalist and constructivist scientific approaches to resilience. Given the diversification of scientific
approaches, resilience cannot operate as a theoretical model or unifying paradigm (Mummery &

Mummery, 2019). Yet, as a metaphor resilience provides a sound basis for reconciling contrasting scientific approaches, mainly because of its heterogeneity and high level of abstraction (Thorén, 2014). Intellectually, the reconciling of naturalism and constructivism implies an appreciation of diverse scientific vocabularies, many visions of what counts as scientific knowledge, other approaches' scientific worlds, a certain embracing (which includes making manifest) of the tensions between the contrasting types of science, and creating spaces for constructive contestation (Pfeffer & Georgiadou, 2019). Thereby, new resilience perspectives may develop. New questions may be posed (or new answers to long-standing questions may be provided). The resilience trap – typically marked by the promotion of adaptive strategies that reify responses and corresponding power structures in the short-term – may be avoided (via challenging current assumptions underpinning resilience research). Current adaptation and transformation and transformational adaptation approaches may be further refined. And much-needed new ways of scientific thinking and possibilities may be opened in resilience research, beyond old conceptualizations and modes of analyses (cf. Fazey et al, 2018). These developments ask for new collaboration frameworks and platforms that empower stakeholders to bring both their resilience research questions and their assets to the table to collectively explore and define potential futures from the perspective of all present worldviews.

A second theme for future resilience research comes with a change in political environment, in which the legitimacy of adaptive, transformative, and transformational adaptive responses to climate change is constantly contested. Anthropogenic climate change comes with a political-administrative crisis, which manifests itself in the form of a legitimacy crisis, authority crisis (including the crisis of scientific authority), crisis of democracy, a crisis of human rights, a crisis of modernity (Swyngedouw, 2011; Blühdorn, 2013; Fischer, 2017; Ossewaarde, 2018; Stegemann & Ossewaarde, 2018; Dryzek & Pickering, 2019). Crisis and the ability to absorb changes and shocks has been widely constructed as the new normal (Hilhorst, 2018). In an increasingly toxic political environment (marked by climate change denial, anti-immigration policies, and nationalist protectionism) adaptive and transformative resilience and transformational adaptation may be expressed and contested in

manifold ways. For instance, on the one hand, environmental protest movements are stakeholders
that develop a leverage required to transform established systems (such as energy systems) and their
governance arrangements. On the other hand, agents who hold power thanks to such arrangements
typically use tactics of repression and criminalization, particularly in the extractive sectors of the Global
South (Szablowski & Campbell, 2019). New research questions emerge on the one hand from
polarization and the exercise of (il)legitimate power in the governing of and for resilience to climate
change. This is the question of how the adaptation and metamorphosis of systems under pressures of
climate change comes with power inequalities, polarization, injustice, battle for resources, democratic
deficits and post-democratic tendencies, climate change denial tactics, attacks on legal rights, and the
resilient governance of unsustainability. To put it in more positive terms, urgent questions concern the
meanings of transformation, the theorization of transformation in terms of just resilience, the linkage
of resilience to sustainable futures, the development of a transformation agenda in participative,
proactive and deliberative ways, and the comparison of different administrative capacities and new
governance arrangements that explain differences in system adaptation and reconfiguration (cf.
Blühdorn, 2013; Fischer, 2017; Davoudi, 2018; Köhler et al, 2019; Mummery & Mummery, 2019).
A third promising topic for future resilience research concerns the relationship between
adaptive resilience and transformative resilience and transformational adaptation in the reactive and
proactive governance responses to anthropogenic climate change (Clément & Rivera, 2017). In the
coming decade, questions like how adaptive and transformative resilience to climate change is
strengthened or weakened; how the current performance of systems when it comes to responding to
possible disturbance (for instance, through the use of monitoring systems) can be better understood;
how unjust resilience can be disabled (and therewith 'positive vulnerability' can be increased to
generate beneficial transformation (cf. Gallopin, 2006); and how transformational adaptation
manifests itself (how multiple adaptations may lead to transformational adaptation and what are the
tipping points for igniting transformation), become urgent ones for resilience research (Grove &
Chandler, 2017; Glaser et al, 2018). The notion of 'tentative governance' appears particularly relevant
in the context of transformational politics, when it comes to phasing out systems and weakening
adaptive resilience. Tentative governance is marked by interventions that are designed as preliminary
rather than as persistent, for purposes of probing and learning rather than for stipulating definite
targets or fixating existing systems and their underlying assumptions (Kuhlmann, Stegmaier & Konrad,
2019). It is likely that stakeholder engagement (including resistance) in transformational politics and
tentative governance varies, and manifests itself differently, across different policy fields. For instance,
the sustainable energy transformation may include multi-layer governance challenges, many pro-
active stakeholders, new investment opportunities and job opportunities. Given that multiple public
and private actors are responsible for the performance of different parts of a system, tentative
governance comes with transformational adaptations that must be arranged. Hence arises the
question which adaptations allow for transformation? In contrast with the sustainable energy
transformation, sea level rise and the disruption and relocation of coastal cities may trigger a more
limited transformative politics, despite inevitable transformation of systems due to shocks and
disturbances (metamorphosis). Yet, in the coming decade, transformational politics and tentative
governance – including anthropogenic topics like population displacement, privatization of climate
adaptation, conflict organized around scarce resources (like water resources), intergenerational
environmental conflict, and the closing of old infrastructures that are too costly to maintain – becomes
a more urgent research topic.
A fourth topic for future resilience research concerns the relationship between phasing out of
unsustainable systems and societal transformations. The sustainable energy transformation is a most
obvious phasing out of old systems (like coal energy systems) and change of worldviews, middle class
consumerism, lifestyles, etc. towards new energy systems, given that burning fossil fuels has such a
major impact on climate change. Adaptative and transformational responses to climate change are
intermingled with responses to many societal and ecological developments. A response like
investment in transportation systems that aims to address increasing transportation demand must
accordingly include possible climate change impacts. In the Anthropocene epoch, systems typically

face pressures to change, to establish new (less unsustainable) interactions between society and ecology. Pressures on existing systems not only emerge from ecological adversity, over-exploitation, resource depletion, etc., but particularly from counter-discourses and new ways of thinking, new lifestyles, and new contestations (like the Fridays for Future, the Anti-Mining, the Transition Towns, Black Lives Matter, and Degrowth movements) that increase the positive vulnerability of undesirable systems (Bergmann & Ossewaarde, 2020). At the same time, anthropogenic climate change comes with the development of a multi-trillion market of the emerging green economy, which proves new climate investment opportunities. Given such societal pressures and opportunities, new research topics include the governing and accelerating of the decline of existing systems and their adaptive cycles (Stegmaier, Visser & Kuhlmann, 2014; Hoffmann, Weyer & Longen, 2017; Stegmaier, Visser & Kuhlmann, 2020); the particular circumstances in which accelerations can manifest themselves; the identification of, and coping with, uncertainties in processes of adaptation and transformation and transformational adaptation; and the construction of new incentive structures, for accelerating sustainable transformation (cf. Clément & Rivera, 2017; Warmink et al, 2017; Köhler et al, 2019). This branch of discontinuation research assumes that technologies influence socio-ecological systems. Some technologies threaten resilience to climate change, while others enhance it (Smith & Stirling 2010). Such research informs that political objectives like drastic reduction of $CO_2$ emissions (as can be found in the European Green Deal (2019) will hardly be achieved by using single cleaner (green) technologies alone, but structural system metamorphosis is needed to qualitatively alter established systems (Vögele, Kunz, Rübbelke & Stahlke 2018; Rogge & Johnston, 2017; Stegmaier 2019). One of the challenges for the coming decade is to reverse the negative, alarmist, catastrophic, apocalyptic or paralyzing image of climate change: transformational adaptation comes with stakeholders taking a pro-active and positive view on climate change and on positive vulnerability, with new opportunities emerging from responses to climate change. How can climate change and vulnerability of established (and typically unsustainable) systems be regarded as an opportunity rather than as a risk in the governance of transformational adaptation to climate change?

A fifth theme for future resilience research concerns the role of environmental, energy and
climate justice in theorizing, modeling, interpreting, and explaining resilience to climate change (cf.
Skillington, 2015; Fazey et al, 2018; Mummery & Mummery, 2019). For future research, theories of
environmental justice, energy justice and climate justice can be conducive to helping furthering
comprehension of adaptive and transformative resilience and transformational adaptation. How can
justice claims be made more responsive to newly unfolding ecological and societal circumstances and
uncertainties? How can principles of equity, fairness and access to resources and services be secured
in a toxic political environment? And how can – in the problematic context of climate-induced
migration and a political environment marked by anti-immigration policies – the wellbeing of migrants
be ensured? Theories of environmental, energy and climate justice are also highly relevant for
developing understanding of how adaptive and transformative resilience and transformational
adaptation are perceived and experienced in everyday life by different stakeholders that face
anthropogenic challenges. Constructivist enquiry into perceptions, experiences and prioritizations of
resilience constructs is a promising topic for future resilience research. In this regard, insurance
decisions of citizens against the risks associated with climate extremes can gain further research
attention. As addressed by O'Hare et al. (2016), citizens are faced with an increasing responsibility to
make decisions to 'insure' themselves and their assets against the possible damages of climate change.
Such decisions can have diverse justice implications in different political and economic contexts that
influence how citizens perceive, experience, and prioritize climate risks. Similarly, the cross-sectional
dimensions of justice, particularly gender and racial relations, is becoming increasingly relevant and
yet challenging to understand and integrate into climate justice (Terry, 2009), and energy justice
(Feenstra and Özerol, 2018) frameworks. And in the Global South, addressing issues of corruption,
violence, poverty and lack of access to resources (and violent battles for resources) and services (like
education and sanitation) may have a higher priority than global environmental considerations (Köhler
et al, 2019).
A sixth theme for future resilience research comes with a changing (geo)technological
environment, that is, the so-called 'AI revolution' in the making. Given worldwide investments and top-
down AI strategies that global governance actors and national governments have recently published,
AI will most plausibly become a major force that shapes resilience to climate change by means of
monitoring, forecasting and learning. A relevant example of big data is the G-Earth Engine and the vast
amount of satellite imagery made available by space agencies, which opens up an unprecedented
dataset of satellite images for scientific research. Such extensive datasets, marked by high spatial and
temporal resolution, are essential for monitoring a changing earth system. In the past decade,
resilience discourses have increasingly incorporated phenomena like big data, AI, cybersecurity and
smart city. In the coming decade, resilience discourses may increasingly become algorithmic
technology discourses. New interplays between automation, (un)sustainability, and adapting and
transforming systems trigger new questions for future resilience research (cf. Köhler et al, 2019). For
instance, in the near future, not only the number of climate disasters is expected to rise. Also the data
– satellite data, drone data, sensor data, social media data, volunteer geographic information (VGI)
data, Internet of Things data, etc. – available on such disasters is expected to increase in size and
resolution, amounting to vast volumes of climate disaster data. However, AI, due to the unstructured
nature or coverage of input data, may omit those phenomena, places and social groups that are not
present in the data (Hoefsloot et al. 2019). Alternative ways of knowing can refine or contribute
complementary insights to the precise measurements and data gaps (Pfeffer and Georgiadou 2019).
New research questions for naturalist and constructivist research emerge from challenges of
organizing big data and how to make it available and usable, given the variety of public and private
stakeholders, workflows and incentive structures involved in the (social) construction of big data
(Wright, 2016). How can AI be augmented with alternative ways of knowing to strengthen
adaptive/transformative resilience? How to incorporate the socio-spatial dimension in resilience
research, to pronounce the different capabilities of different groups and places? And what role can AI
play in creating a dialogue between the naturalist and constructivist resilience research? In the coming
years, AI tools – mainly tracking (for instance, tracking of deforestation tracking or energy/water
consumption) and machine learning techniques – are expected to be widely used. Among other things,
for detecting and predicting how climate disasters probably develop, for locating areas or communities
at risk, for analyzing the consequences of climate disasters, and for assisting in climate disaster
responses. Working with AI for purposes of learning from data – for instance, via the use of data mining
or deep learning techniques for dissecting patterns in satellite images – comes with the design of
procedures for data analytics, forecasting and intervention (Rodríguez-González, Zanin & Menasalvas-
Ruiz, 2019) and requires domain and local knowledge as well as a dialogue between naturalist and
constructivist researchers. In contrast to the official national statistics of the past, which diffused
societal controversies, big data analytics create myriad parallel realities, stand in the way of achieving
a minimal consensus about basic facts and amplify controversies. A recent example where AI and
alternative ways of knowledge came together is the resilient settlement program led by UN HABITAT
which brought together a multitude of actors (policy, private, academic, community organizations) and
data and algorithms and local knowledges to identify settlements at risks. In sum, next to
technologization of resilience discourses, social processes of big data construction, the inclusion and
exclusion of diverse stakeholders, the embeddedness of AI in everyday practices, the various uses of
AI in the exploitation of data, fair, transparent and accountable (FAT) AI, as well as the integration and
inclusion of alternative knowledges are promising fields of resilience research.
In the coming decade, several AI challenges are most likely to increasingly come to the fore in
resilience research. First, monitoring systems (for instance, monitoring the status and behavior of
infrastructure or human settlement dynamics) that incorporate machine learning make that systems
are automatically checked rather than regularly inspected by experts. When AI is integrated with
knowledge of how systems work, expertise is outsourced to AI, which implies that expert knowledge
may get lost or become obsolete. Moreover, AI classifications may have unintended consequences for
certain places or communities. For example, by labelling areas at risks, property prices may go down
or insurance agencies are not willing to provide an insurance certificate. Second, the digitalization of
SES makes systems vulnerable to, for instance, breakdowns, power outages and cyberattacks – hence
resilience strategies and digital strategies are intertwined (Wessel, 2019). 'Digital resilience' has
recently become a key concept in resilience research that refers to strengthening resilience of digital
systems to potential cyberattacks, including the adaptive capacity to respond to such attacks (Wright,
2016). The making of digital resilience typically implies bringing in tech firms for the protection of SES,
whose algorithms are typically opaque. Third, because of the reliance on AI and associated data, other
realities are neglected, excluding certain places or communities from digital resilience strategies.
Fourth, AI systems facilitate governing at a distance, with governing becoming more invisible and
possibly unaccountable. For instance, when disaster management (for instance, in the context of an
extreme weather event) becomes 'digital humanitarianism', the distance between the saviors and
survivors becomes big, with survivors becoming reified abstract entities that inspire limited empathy.
In fact, survivors are confronted with the risks of AI systems, in terms of privacy breaches and identity
frauds. In other words, while AI is expected to become a key theme in resilience research, a promising
topic for future resilience research concerns the challenge of uncovering resilience traps and
neutralizing the ecological and societal damage and injustice done through the reinforcement of AI
technologies in governance processes like digitally-based service provision or humanitarian
interventions in the Global South.


**5. Conclusion**

In the social sciences, resilience to climate change is a concept that is incorporated in different
theoretical approaches that are linked to contrasting scientific approaches. Holling originally
reinterpreted and incorporated the good old notion of resilience in his SES approach, which was then
picked up by naturalist scientists who incorporated Holling's reinterpretation of resilience in their own
cybernetic complexity theory. The naturalist complexity theoretic approach to resilience as system
adaption to climate change was dominant in the social sciences, until the ecological and political (and
increasingly also the technological) context of resilience research changed. When a decade ago actors
at global, national and local governance levels drafted their resilience policies in the wake of socio-
ecological catastrophes, financial crises, climate crises, pandemics, governance failures, and the
breakdown of infrastructures, constructivist approaches developed to take resilience research far
beyond complexity theory and associated methods. And it introduced a variety of new concepts for
resilience research, such as the resilience discourse, myth of resilience, just resilience, resilience trap,
transformative resilience, and transformational adaptation. Resilience cannot operate as a unifying
paradigm, but it can facilitate the reconciliation of naturalism and constructivism. Thereby, the two
contrasting scientific approaches can provide a liberating perspective on each other (without the one
repressing the other) and brought into a theory-energizing tension with each other. Such reconciling –
igniting theory-energizing tension – is needed for reimagining resilience to climate change and for
specifying how new political-administrative institutions (including panarchical self-organization) and
practices can respond in legitimate ways (taken justice and vulnerability considerations into account)
to the challenges of climate change, in different ecological, political and technological contexts (cf.
Johnsson et al., 2018).

Given recent developments in the social sciences, the key resilience issue concerns the political

response in the form of adaptation, transformation, and transformational adaptation in newly
unfolding political, ecological, and technological environments. The six resilience themes for the
coming decade that this paper has identified are all connected to the issue of the political-
administrative response to the challenges that come with anthropogenic climate change. A first theme
concerns the reconciliation of naturalism and constructivism, to be able to move beyond established
assumptions, theories, concepts, and modes of analysis; and to trigger new imaginations to be able to
create new, theory-rich, resilience perspectives. A second theme is the legitimacy of the political
response in a toxic political environment, in which top-down and bottom up responses, including new
governance arrangements and system reconfigurations, may suffer from legitimacy deficits. A third
theme is how, in a toxic political environment, adaptation, transformation and transformational
adaptation can be materialized; and under which conditions such governance responses are sufficient
for addressing climate change challenges. A fourth theme is how systems are under pressure due to
climate change, ultimately igniting a phasing out of systems and a departure from environment-
unfriendly consumerist lifestyles, values, and assumptions. A fifth theme is how governance responses
can be made legitimate, by incorporating considerations of environmental and climate and energy
justice, thereby strictly connecting resilience to justice considerations. A sixth theme is how new
technologies (mainly AI) come to intermingle with resilience: what is the role of such technologies and
giant tech oligarchies like Google and Amazon in political-administrative responses to challenges that
come with climate change? And, correspondingly, what are the undesired consequences that come
with AI and giant tech firms, when it comes to responding to climate change. How does AI enact
existing power structures, thereby reinforcing resilience traps?

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
