# Peer review of "Review article: Towards a context-driven research: a state-of-the-art"

_Natural Hazards and Earth System Sciences, 2020_

## Referee Comment (RC1) · Anonymous Referee #1 · 10 May 2020

This paper presents a discussion on different ontological and epistemological strands underpinning resilience research. It aims to shed light on the consequences of taking naturalist approach as opposed to a constructivist approach to studying social-ecological resilience by reviewing the existing literature on the topic. It then proposes to define a research agenda for future resilience research, focusing on the particular concepts of adaptive and transformative change.

Overall, this paper is an interesting addition to the existing literature as it aims to reflect on the assumptions driving resilience researchers and thereby improve future conceptualisation of the research. Furthermore, its structure is logical and the authors

present a gradual approach to presenting the core concepts of the paper, that is naturalist and constructivist resilience research approaches, and adaptive and transformative change. However, I found several significant issues with the paper which will need to be tackled before publication. These concerns mostly the first chapters, since I generally find the last one on the proposed research agenda clear.

First, I wonder if the scope of the paper fits the journal. The paper focuses on the philosophical underpinning of resilience research, rather than the more natural science and policy-focused nature of NHESS. One may argue that the paper could be of interest to natural scientists working on resilience to climate change hazards, but the article is difficult to follow (see comments below), except the last section on the proposed research agenda.

Second, the writing style makes the article very difficult to follow. This is probably because the paper uses long sentences and many different terms, often using concepts and complex terms to explain already complex terms. To take an example, in Section 2.1. which aims to explain naturalist resilience, references are made to "logical positivism", "cybernetic", "complexity theoretic orientation" without defining these additional concepts. It makes it difficult for the reader to capture the main conveyed idea. Many terms would need to be better explained, fewer ideas perhaps presented and certainly more systematically illustrated using concrete examples. A thorough check on the grammar is also necessary. I take two examples, but there are others throughout the text:

- Line 206-207 "Ecological and societal catastrophes... manifest such no-liberalised resilience that is divorced from concerns of justice". I do not understand this sentence, which might be due to a grammatical error. - Line 389-392 "transformational adaptation means...via stakeholder participation" does not explain anything to me: what is an "energy establishment"? what does it mean to transform an "establishment" into an "agent of change"? In what ways is "stakeholder participation" contributing to a transformational change, since this is already practiced in more incremental, adaptive

change?

Third, the paper could be more clearly framed from the outset. For example, I only realised that the authors focus on resilience research carried out in the social sciences after the introduction. It would also be useful to clearly focus on one issue (e.g. resilience to climate change) which the authors appear to suggest sometime later in the article.

Fourth, the paper focuses very much on the very early SES research (by Holling etc) or on very recent literature (from 2014-2015). While I appreciate that the focus of the review article is on recent debates, I would recommend a more thorough presentation of how the concept of resilience was debated in the 1990s and especially in the 2000s in the environmental sustainability field and the growing field of research on global environmental change. Significant debate for example occurred in the 2000s to define resilience, as opposed to the then prominent concept of vulnerability, e.g. researchers such as Gallopín, Duit, Brooks, Dieaz, Adger, Smit. They will shed further light on the origin of the idea that resilience transposed to society is a neoliberal concept that puts more emphasis on individual responsibility and ignore the social and political factors leading to inequity in face of climate hazards. This historical background does not need to be long, but it should at least acknowledge this literature, perhaps using it in Section 2 when it sets out to explore the emergence of the concept of resilience.

Fifth, and related to the above comment, given the key concepts of the paper focusing on resilience and adaptive and transformative change, I am curious as to why the concept of adaptive capacity was not, at least, acknowledged, if not used to inform the discussion. It seems to be a component of resilience thinking, which has provided an analytical framework for much governance research on global environmental change. How is that strand of literature linked to the growing interest in adaptive and transformative change? In what ways do the later concepts offer fresh and new insights?

Sixth, the particular focus on ABM and AI are welcome and interesting, but they need

to be better justified. Why are these examples of tools used to research resilience mentioned more than others? More generally, the discussions on Section 3 could provide more concrete examples of the methodological implications of taking one approach or another.

Finally, the readability of the article could be improved with the use of figures and tables to lighten up the text, for example to present definitions or how the core concepts of the paper relate to each other.
* * *

---

## Referee Comment (RC2) · Md. Shibly Sadik (Referee) · 16 May 2020

The paper brings a new way of thinking resilience. It discussed the current discourses and approaches of resilience science from a different perspective – naturalist and constructivist. Arguments were nicely presented with necessary discussion and clarification in most of the cases. However, there are some issues and limitation appeared to me which should be resolved or addressed before publication. First, the introduction chapter was difficult to follow. It appears that it suddenly jumped to introducing the content of the paper after mentioning the aim of the research. It is difficult to follow whether it is a part of the background review or methodology.

[Figure]

Besides, refereeing to published research is necessary for a few places (please see the attachment for detail). Second, at several locations, SES is correlated to the debate between naturalist and constructivist perspectives of resilience which is interesting. However, given the context of the research, especially the discussion on adaptive resilience and transformative resilience, it would be appropriate if the author tried to bring the Panarchy theory (by Gunderson and Holling 2003) in the discussion, it would be more interesting and appropriate. It would be interesting to know whether his statement "adaptive resilience obstructs transformative resilience" aligns or conflicts with the Panarchy theory of adaptive cycle and resilience building. Third, the author tried to generalize that the application of AI strengthens adaptive resilience and weakens transformative resilience. The reference he provided to justify his argument (i.e. Khakurel et. al. 2018) does not entirely support his comment. To establish his comment/judgment he needs more examples and arguments. However, it cannot be generalized for all cases. One can argue that the use of AI can build transformative resilience. For example, the capacity for anticipating future events is an element of transformative resilience. AI can help us in anticipating future events more accurately. Thus, AI helps to build transformative resilience. Fourth, section 2 – diversification of resilience research, it is not easy to follow the direction of the discussion. If the authors explicitly mention the direction of the discussion in the beginning and the key messages at the end, it would guide the readers. Fifth, at several places the authors need to validate his statement or argument with appropriate references. For example – In lines 196-199, the authors mentioned that when government failed to provide water supply, it advocated for community-based water management in East Africa. Such statement should be validated with appropriate references. In lines 210-212, authors mentioned that hurricane Katrina as a racial flood that brought back colonial patterns of racism, slavery, and vulnerability which must be validated by appropriate reference. Sixth, the authors used many jargons some of them might be very common in social and political science and a few of them are new, for example, "climate gentrification", "climate apartheid", "transfiguration", etc. It would be helpful for the reader if these jargons are briefly defined in the text for the reader.

Please also note the supplement to this comment:
https://www.nat-hazards-earth-syst-sci-discuss.net/nhess-2020-90/nhess-2020-90-RC2-supplement.pdf

**Supplement:**

[revised manuscript text omitted]

---

## Author Response (AR1)

Dear referee,

Thank you very kindly for your comments on our paper and for your critical and constructive feedback that will enable us to improve it. You give us seven points of feedback. We have worked with your feedback in the following way.

1. You mention that the first sections (up to the proposed research agenda section) are difficult to follow for natural scientists and policy-focused scientists. In many ways this is the core of your feedback that also informs some of the other points of your feedback. We will take this point of feedback seriously, keeping your advice in mind (given in the points below) and revise the article – particularly up to the research agenda section. Specifically, we will ensure that the article's writing style, formulations, line of argumentation, conceptualization, choice of words etc. are easy to follow for a broader audience.

   We have actively rewritten the article (as can be seen in the track and trace version) in line with this comment. We have done our best to make our article readable and easy to follow.

2. The second point of your feedback stresses that the writing style is difficult to follow, which is linked to your first point. In line with your advice, we will improve the writing style, replacing complex terms and shortening sentences. We will also ensure that concepts are clearly defined, and better explained, illustrated and concretized, without introducing too many concepts. Further, we will have a careful look at the grammar and clarity of sentences. You give examples of unclear sentences, which we will address with care, and we will go through each sentence to ensure clarity throughout the paper.

   In our revision, we have actively worked with this comment. We have improved our writing style, actively shortened our sentences, and have more clearly defined and explained our concepts. In our revised manuscript, we have actively worked to improve our clarity.

3. Your third comment refers to the framing of the article. You give the useful suggestion that the article should be framed as resilience research in the social sciences and with the focus on climate change from the very beginning, in the introduction section (and in the abstract). We will implement this suggestion by the first author of our paper.

In our revision, we have actively worked with this comment. We have rewritten our
introduction section and abstract. We have now stressed from the very beginning that the
paper concerns resilience research in the social sciences and with a focus on climate change.

4. Your fourth comment refers to providing more historical background of the SES notion of
resilience. This should include how it was debated in the 1990s and 2000s in the environmental
sustainability field and the growing field of research on global environmental change. And as
you suggest, we will emphasize the debates that occurred in the 2000s, to define resilience as
opposed to vulnerability. Thank you for suggesting relevant references for describing and
acknowledging this background.  We take this fourth comment at heart and we will include
the discussion on the historical background, along the lines that you suggest.

We have actively worked with this comment, including some of the vulnerability literature that
the reviewer suggested. We have included the resilience as opposed to vulnerability argument
and included it in our line of argumentation.

5. You wonder why we do not emphasize adaptive capacity in our discussion, given that adaptive
capacity has provided an analytical framework for much governance research on global
environmental change. You suggest to link that strand of literature in our discussion of adaptive
and transformative change. From our side, there were no principal reasons for omitting that
body-of-literature in our discussion. We take your advice at heart and link up with that body of
literature. In our revised article we will specifically work with the questions that you provided,
namely, 'how is that strand of literature linked to the growing interest in adaptive and
transformative change? In what ways do the later concepts offer fresh and new insights?' In our
revised article we will specifically work with the questions that you provided, namely, 'how is that
strand of literature linked to the growing interest in adaptive and transformative change? In what
ways do the later concepts offer fresh and new insights?' Amongst other things, we will refer to
Ziervogel, G., Cowen, A., & Ziniades, J. (2016). Moving from adaptive to transformative capacity:
Building foundations for inclusive, thriving, and regenerative urban settlements. *Sustainability*
*(Switzerland)*, *8*(9). https://doi.org/10.3390/su8090955

In our revised manuscript we have worked actively with this comment and also included the
Ziervogel, Cowen and Ziniades article, which was indeed very helpful and relevant for us.

6. You stress that the particular focus on ABM and AI need to be better justified, and explained why they are mentioned more than others. And you stress that 'the discussions on Section 3 could provide more concrete examples of the methodological implications of taking one approach or another.' For us this is a comment and advise that we take seriously. We will work with the comment, doing our best to improve our justification and concretization. The focus on ABM we will justify more strictly as a typical and frequently used approach that we encounter in contemporary naturalist resilience research. We will mention other naturalist approaches that are found in naturalist resilience research. And we will better justify AI in terms of the so-called 'AI revolution' that is currently shaped by governance actors. And this 'AI revolution' has implications for both socio-ecological systems and for resilience research.

In our revised manuscript we have worked actively with this comment, working to better embed ABM and AI in our article.

7. You suggest to improve the readability of the article with the use of figures and tables, for example to present definitions or how the core concepts of the paper relate to each other. In our revised article we will take this useful suggestion into consideration, as part of the general effort to improve readability of the article. Our plan is to develop a figure that visualizes how the core concepts of the paper relate to each other.

We have actively thought about this comment. After internal discussions amongst ourselves we have decided to leave out the visualization as we believe that it does not fit nicely with the style of our article. If the editor would insist on a visualization, we can of course include it, for instance in the form of a table.

Dear referee,

Thank you very kindly for your comments on our paper and for your critical and constructive feedback that will enable us to improve our paper. You give us six points of feedback. We wish to work with your feedback in the following way.

1. You mention that the introduction section is difficult to follow and unclear. You stress that in the introduction section it is difficult to follow whether it is a part of the background review or methodology. We take your comment at heart and revise the introduction section, in line with your comments.

   We have actively worked with this comment. We have rewritten the introduction section (as well as the abstract) in line with your comments.

2. You give us the advice to bring the panarchy theory in our discussion on adaptive and transformative resilience, particularly to find out whether "adaptive resilience obstructs transformative resilience" aligns or conflicts with the Panarchy theory of adaptive cycle and resilience building. We find this an interesting and relevant advice that we will follow in our revised manuscript. We will add the discussion on the panarchy theory to the discussion on adapative and transformative resilience.

   We have actively worked with this comment. We have included panarchy theory in our discussion.

3. You stress that we generalize too easily that the application of AI strengthens adaptive resilience and weakens transformative resilience; and that we need more examples and arguments for this. You stress that it cannot be generalized for all cases; and that AI can also help to build transformative resilience, given that the capacity for anticipating future events is an element of transformative resilience. We find your comment very relevant and will work with your comment in our revised manuscript, rethinking our argument. In revising our manuscript, we will include more concrete examples, which we will discuss amongst the co-authors of our article.

   We have actively worked with this comment. We have included more concrete examples. And we have emphasized that AI can also help to build transformative resilience.

4. You stress that section 2 needs more direction in the discussion. We take your comment at hear. We will revise this section, to ensure structure and readability and guidance for the reader, being explicit in the point that we seek to make. We will ensure that the article (its writing style, its formulations, its line of argumentation, its conceptualizations, its choice of words etc.) is easy to follow for a broader audience. In line with your advice, we will revise the writing style, replacing complex terms and making shorter sentences. And in line with your feedback, we revise the article to ensure that concepts will be more clearly defined and better explained and illustrated and concretized, without introducing too many concepts.  Also, in line with your feedback, we will have a careful look at the grammar and clarity of sentences. And our plan is to develop a figure that visualizes how the core concepts of the paper relate to each other.

We have actively revised section 2 (as can be seen in the trach and changes), in line with your advice. We have actively worked on our writing style, to make our article more readable.

5.  You mention that in some places we fail to include appropriate references in our discussion; and you give examples of this. We will revise the paper with your comment in making, making sure that we make the appropriate references.

We have actively worked with this comment. We have included the examples you have given us and we have used the references, for which we are thankful.

6.  You mention that we frequently introduce jargon that we leave undefined, and you give examples of this. With your comment in mind, we will revise the article, making sure that if we introduce jargon or concepts, we describe them accurately.

[revised manuscript text omitted]

in the social sciences. Thereby, we seek to resilience research.

, ultimately, we seek to do so

Naturalism and constructivism are presented as two (social) scientific approaches underpinned by  different epistemological and ontological commitments. It is suggested that  to advance into resilience in the context of climate change resilience research could be raised to a next level if these two different approaches meet and interact, we argue, these two approaches need to be bridged. To this end, we reconstruct contemporary debates in that particular field of studies and distil recurrent research topics that divide social scientists. The issues of adaption and transformation in the context of severe disturbances or shocks that come with climate change (such as hurricanes, floods, drought, and heatwaves) appear to be such divisive topics. Second, contemporary key issues of debate in naturalist and constructivist approaches to resilience to climate change research are identified. Ultimately, in the social sciences, naturalist and constructivist resilience research clashes on the issue of system adaptation and transformation in a context of severe disturbances or shocks that come with climate change (, such as hurricanes, floods, drought and heatwaves). The tension between adaptation and transformation has, amongst other things, implications for social scientific enquiry into the sustainable energy transformation, the relationship of resilience research to sustainability discourses, and the response of resilience research to new political and technological circumstances. ThirdFinally, naturalist and constructivist directions, as well as possible cross-fertilizations of these two currents, for future resilience research are identified., including the bridging of naturalist and constructivist resilience research. We point out that future resilience research in the social sciences – that is, the types of questions raised, theoretical frameworks and modes of analysis – will also be determined by changing conditions (ecological, political and socioeconomic). We emphasize , with an emphasis on the likely impact of changing conditions – particularly in ecological, political and technological dimensions – on the questioning, theorizing, and modes of analysis in resilience research.

**2. The diversification of resilience research**

One of the earliest appearances of the term resilience – in European literature at least – seems to have been in one of Aesop's fables, namely, that of The Oak Tree and the Reeds. According to one of the versions of that story, the Oak Tree becomes uprooted during a storm while its fellow reeds survive it.

In a conversation, the Oak Tree expresses its bewilderment that the fragile reeds were able to resist such a mighty storm while it succumbed. The reeds reply that it is precisely their non-resistance that saved them. Through their capacity to bend, they moved with the direction of the wind (which thus did not break them) and rose again when the storm was gone. They were flexible enough. The reeds

'bounce' back and are thus 'resilient'. Indeed, the English word resilience derives from Latin (*resilire*), which generally meant rebounding. This Latin word can be found in the writings of Seneca the Elder,

Pliny the Elder, Ovid, Cicero, and Livy; Lucretius' to rebound is also the sense in which *resilire* is used by Cicero in his *Orations* On the Nature of Things and Cicero's *Orations* (Alexander, 2013; Pizzo, 2015).

The term also appears in Lucretius' *On the Nature of Things*, where it denotes 'being forced back by a resisting surface […] with reference to the action on Nature' (Pizzo, 2015). Along this line, nature compels all things to 'spring off'. Despite the various meanings attributed to the term, the connotation attached to *resilire* was commonly that of rebounding. Up to the early nineteenth century, this was the predominant understanding of resilience in common language and imagination. A slight shift appeared when engineers started to use the term to refer to , until engineers come to employ the term. In engineering, resilience refers to , until engineers come to employ the term to describe the properties and capacities of materials and the capacity of materials to absorb stresses tensions and release energy, and recover their original forms, without breaking or disfiguration ing, after undergoing some external shock or disturbance (, such as an extreme weather conditions event) (Estêvão, Calado

& Capucha, 2017; Bergström, 2018; Davoudi, 2018). In the 1950s, psychologists re-adapted the common sense of the term to mental health and used it to turn to resilience to analyze study the coping mechanisms of concentration camp survivors. L; later, the concept is used to study all sorts of trauma, misfortune, adversity, stress stress, and mental recovery (Bourbeau, 2015; Estêvão, Calado & Capucha,

2017; Bergström, 2018; Schwartz, 2018). In the 1970s, the ecologist C.S. Holling (1973: 14) redefines resilience as 'a measure of the persistence of systems and their ability to absorb change and disturbance.' Thus understood, resilience is widely conceived as the opposite of vulnerability, which is defined as the inability to absorb change and disturbance (Gallopin, 2006; Miller et al, 2010) - (for instance, a coastal system that is vulnerable to  accelerated sea-level rise is not resilient enough (Smit, Goosen & Hulsbergen, 1998). In such discourses, greater  resilience means  becoming less vulnerable to change and shocks.  That said, Aa system can still be vulnerable to other changes while being resilient in other respects  (Gallopin, 2006).

Holling incorporates resilience in a socio-ecological systems (SES) approach to analyze the stability and strength of ecological systems, which are constituted by the interaction between natural ecosystems and human societies ~~assemblages as conditioned by, and conditioning, societies. Holling emphasizes the relationship and interaction between ecological systems and social systems. Hence, in Holling's work, resilience has a relational and systemic focus in scientific enquiries into how nature and society interact — a line of enquiry that brings the social sciences, the natural sciences and engineering together in an overarching SES frameworkOne could say today that a ubiquitous concept like resilience expresses a 'governmental philosophy of nature and society' (Walker & Cooper, 2011: 145), the ability par excellence to survive conflict and crisis.~~

In the social sciences, resilience research that has emerged from Holling's SES approach has developed along two different lines, which can be called naturalist and constructivist, respectively (Miller et al, 2010).  These two currents of research have different focuses, raise different questions and have recourse to different methods. The naturalist line of research is indebted to the accepted methods and assumptions of the natural sciences. It has a predilection for mathematical and simulation models. Social scientists dealing with resilience to climate change research questions consider resilience as a property of a system, which can be (made) weak or strong.

manner of the natural sciences. In resilience research, naturalist scientists they typically model s, with the world being modelled as consisting of physical properties that can be objectively studied  resembling atoms, mass, molecules, cells, DNA, etc. (Aiken, 2006; Floridi, 2017).  – Moreover, history and culture (in the sociological sense of the term) cannot be integrated in the various models. Resilience as a system property is an objective measure of the dynamic equilibrium, stability, strength, or survivability of a socio-ecological system, , including coastal systems, urban systems, forest systems, etc. (Hoekstra, Bredenhoff-Bijlsma & Krol, 2018).

The naturalist approach to problems that arise through climate change can be very useful, especially when both the problem and the solution are quite uncomplicated (and hence are primarily of a technical nature, such as water purification, for instance). The story becomes more complicated when, for instance, attempts to make communities more resilient to climate change overlook the political and cultural reasons why particular groups are more vulnerable to the effects of climate change. Since a model cannot include these reasons, the naturalist social scientist necessarily leaves out factors that are part of the problem and the solution. In so doing, naturalist social scientists may well become unwitting allies of political powers and help to perpetuate status quos. Constructivist social scientists have shown increased interest for resilience research precisely because resilience is a term profusely used by global and national powers during the last two decades.

~~is a type of science that seeks to explain the world in the manner of the natural sciences, with the world being modelled as consisting of physical properties (Aiken, 2006; Floridi, 2017). Resilience is likewise defined as one of the system properties (Hoekstra, Bredenhoff-Bijlsma & Krol, 2018). In naturalist research, resilience is defined as a system property: resilience is an essential measure of the dynamic equilibrium or survivability of a socio-ecological system.By contrast,In the social sciences, cois an anti-naturalist scientific approach that researches phenomena as subjects invested withIn the social sciences, constructivists emphasize that social sciences are fundamentally different from the natural sciences, because social phenomena are fundamentally different from physical properties.invested with a is a type of science that denaturalizes and historicizes, in~~

particular understandings of nature, society and the person, of values, symbols and historical practices (which may not be very rational or just), and power relations.

Constructivists tend to be more critical and politically sensitive. They are generally  more aware of the potential and actual abuse of power. When addressing resilience issues in the context of climate change, they typically express concern for vulnerable communities,

Research topics thus include the

It is more critical and politically sensitive. It typically expresses concern for

(un)equal distribution of environmental burdens, struggles for recognition, claims to participation, and unequal impacts of anthropogenic climate change (Braun, 2014; Yanarella & Levine, 2014;

Skillington, 2015; Sjöstedt, 2015; Weichselgartner & Kelman, 2015; Pizzo, 2015; Lockie, 2016;

Derickson, 2016; Lyster, 2017; Schlosberg, Collins & Niemeyer, 2017; Mummery & Mummery, 2019).

Davoudi (2018:

5) for instance, introduces the notion of 'unjust resilience'. Unjust resilience refers to absorption of changes or disturbance through a systematic neglect of vulnerable groups and marginalized people.

Katrina and the Covid-19 crisis reveal such systematic injustice.

And Glaser et al (2018: 3) refer to 'undesirable resilience', 'bad resilience' and 'wicked resilience'. These are notions that emphasize how resilience may go hand in hand with the enforcement of an undesirable or unjust condition. The resilience of oppressive systems (like tyrannical regimes) that systematically marginalize, discriminate or persecute certain groups are an example of this. to show how, as a construct, the making of resilience to climate change comes with power abuse, domination and injustice. In other words, for the constructivist social scientist, resilience is far from being a neutral property of a neutral system (neutral in the sense of being 'value-free'). Therewith, the theme of anthropogenic climate change in general and the constructivist notion of resilience in particular is placed within wider problematic contexts marked by unequal power relationships.

[revised manuscript text omitted]

**3.2 Transformative resilience and sustainability**

For constructivist scholars, transformative resilience is a post-neoliberal construct that is intertwined with the notion of sustainability. For constructivist scholars, . In constructivist resilience research, the notion of sustainability is transformative. sSustainability is based on the idea that existing systems can be transformed – with respect to social, cultural, political, administrative, economic, technological and environmental factors –, with the right governance interventions and reconfigurations of the ecological and social underpinnings of SES (Pizzo, 2015; Weichselgartner & Kelman, 2015; VanderPlaat,

2016; Ziervogel, Cowen & Ziniades, 2016; Hughes, 2017; Jesse, Heinrichs & Kuchshinrichs, 2019).

Currently, the sustainable energy transformation is no doubt the best example of such a reconfiguration (Park et al, 2012; De Haan & Rotmans, 2018). Fossil energy sources like coal, oil and gas are largely responsible for carbon dioxide emissions, which generate global warming. The sustainable energy transformation, accordingly, is, amongst other things, a response to climate change that is potentially transformative in negating and transcending established (climate unfriendly) energy systems. From the (typically naturalist) perspective of strengthening adaptive 'energy resilience' (Béné et al, 2018: 120; Jesse, Heinrichs & Kuchshinrichs, 2019: 21) – energy systems must adapt to changing environments in which high levels of greenhouse gas emissions comes from burning fossil fuels for electricity, heat and transportation. Energy resilience means that established energy systems can limit the risk of power outage and continue providing reliable energy supplies at stable costs, even in a turbulent ecological and political environment (Wiese, 2016). The notion of energy resilience, as a form of adaptive resilience to climate change, implies that the energy transition, including the use of renewables, can only go via incremental changes and greening of the established economy, to avoid system collapse (Berbés-Blázquez et al, 2017; Schilling, Wyss & Binder, 2018). The middle ground of Transformational adaptation includes this adaptationist notion of energy resilience but aligns it to the scale of desirable ecological and societal changes that are informed by justice considerations and political direction towards less unsustainable futures. Given that established energy systems insufficiently respond to ecological and societal challenges of climate change, transformational adaptation may imply the metamorphosis of energy systems.

[revised manuscript text omitted]

More specifically, strengthened adaptive resilience typically (but not necessarily) may weaken the transformative resilience that is needed for materializing sustainable transformations (Khakurel et al, 2018). In the social sciences, constructivist scientists tend to have a critical view of AI. They do recognize that AI may help building transformative resilience, for instance, when it comes to the phasing out of systems. Yet, from their critical angle, they stress that to make AI serve transformative resilience requires that the domination of giant AI firms is kept in check. And it requires high levels of transparency and stakeholder involvement in how algorithms are designed, built and applied.

In constructivist researches, it is frequently argued that although big data can be openly accessible (like satellite imagery for geospatial and data scientists), big data and AI are often in the hands of giant tech oligarchs like Google, Amazon, Apple, Microsoft, Facebook and Chinese forces (Miller, 2019), that, like the oil barons, are established powers that have a vested interest in the further acceleration and consumption of technological devices (Khakurel et al, 2018). Because of such an oligarchical power structure, AI tends to obstruct transformative resilience, exerting power beyond rule of law and democratic will and understanding (as found in the many recent privacy rights violations, scandals (like the Facebook Cambridge Analytica data scandal (2018), the many Google scandals, etc.), and mistrust of new technologies) (cf. Taddeo & Floridi, 2018; Ossewaarde, 2019).

Moreover, constructivist scholars mention that AI can weaken transformative resilience because we trust too much on the possibility to adapt, and then do not want to change things structurally in a democratic and sustainable way.

[revised manuscript text omitted]

---

## Referee Report (RR1)

**Reviewer Report**

The authors have done a remarkable job at addressing previous reviewer comments and concerns and incorporating their suggestions. Only a few minor edits, as indicated below, are required before I would recommend the article to be accepted.
* * *
L21: system thinking -> system**s** thinking

L69: **has** started to gain? "has" would refer to something that has started recently - but the auhtors refer to the beginning of the new century. I would recommend deleting "has".

L78: It may be interesting for the authors to note a current discussion surrounding the terminology "natural disaster" vs simply "disaster" (see Kelman, 2020).

Kelman, I. (2020). Disaster by Choice: How our actions turn natural hazards into catastrophes. Oxford University Press.

L112: drought -> drought**s**

L169: I would request the authors to remove the example tsunami - as an example of the effects of climate change - unless they can somehow clarify that tsunamis are not    an effect of climate change, rather the impacts of tsunami are likely to worsen because of the effect of climate change on sea-levels. Also, for the other disasters, the authors should be conistent in their use of plural forms - hurricane**s, ... ,** drought**s**

L172: **had** arisen

L175: causation" (Andler, 2014, p. 286)) -> causation" **[**Andler, 2014, p. 286**])**

L176-177: "Most social constructivists do not believe that reality is objective in the naturalist sense (strictly defined) and can thus be fully grasped" -> A bit confusing in my opinion. Do the authors mean to say that because reality is considered to be subjective by most social constructivists - reality **cannot** be fully grasped? If so then, then the authors need to change the sentence to -> ...and thus **cannot** be fully grasped.

L185: the(un)equal -> the (un)equal

L233: "interacting" / Interact has been used two times in the same sentence. Kindly remove one instance of it -> "... diverse  agents that make decisions, interact and learn..." OR "... diverse interacting agents that make decisions,  and learn ...."

L240: micro-macro scale-> micro and macro scale

L250: intervention -> intervention**s**

L261: Avoid back-to-back brackets/paranthesis: "... system property) (cf. Andler, 2014)" **->** "... system

property**;** cf. Andler, 2014)"

L276: governments" -> governments'

L295: do the authors mean to stay rising **sea** levels? Please change accordingly.

L301: very concept reslience -> very concept **of** resilience

L305: **a** unifying paradigm

L307: Resilience to climate change research -> I would say that the following sounds less confusing: **Climate change resilience** research ...

L325: When the authors state "between social scientists" - the question arises - between social scientists and whom? However, if they mean that there is disagreement within the social science community - then I would ask the authors replace "between" with "**among**".

L340: (growth, accumulation, restructuring, and renewal) (Folke, 2006) -> (growth, accumulation, restructuring, and renewal**;** Folke, 2006)

L364: As the Meuse flows through several countries, it is not clear which government the authors are referring to. The authors should try to modify the text in a way that it can easily be grasped by an international audience.

L368: a research -> research

L370: "at the level of land-use" -> does not sound appropriate. Do the authors mean "in terms of land-use" / "with respect to land-use".

L371: The authors may want to be clearer about the location/ geographical context of the research finding. Also "found out that" -> "found that"

L381: Do the authors mean **anthropogenic** shocks?

L390: It is quite a statement to make that structural power is not measurable (arguable/debatable at the least). See:

Brams, S. J. (1968). Measuring the concentration of power in political systems. The American Political Science Review, 62(2), 461-475.

Mayhew Jr, B. H., Gray, L. N., & Richardson, J. T. (1969). Behavioral measurement of operating power structures: Characterizations of asymmetrical interaction. Sociometry, 474-489.

Upong having a fast look at two of the cited references - Lockie (2016) and Howell (2015) -, I have not understood where they state that structural power cannot be measured (I may be mistaking). I would request the authors to double-check such citations, and to be careful that the references are cited in the correct places.

L398: "long term sustainability" -> long-term sustainability

L399: <space> after positions

L426-29: any pollutant that cause global warming is a greenhouse gas. Therefore I would suggest to change the sentence from: "...degradation, water pollution, as well as greenhouse gas emissions and other pollutants that have been causing global warming (Cook et al., 2016)." to ""...degradation, **air and** water pollution, as well as greenhouse gas emissions that have been causing global warming (Cook et al., 2016)."

L442: green technological ? do the authors mean green technologies or green technological innovation

L462: They propose **that** the creation

L468: Is "upcoming" in "upcoming themes" the appropriate word to use? Or is a better terminology would be "evolving themes", "emerging themes", "recurring themes". By the very definition of the word - "upcoming" means something that is about to happen / forthcoming . These research themes have already appeared in the literature, and hence I feel the choice of the word is not appropriate.

L482: Continuing attention -> **continued** attention OR **continuous** attention

L488: inter-, multi- and transdisciplinarity

L489: It  entail**s**

L496: Wilson'**s ;**    consilience should be within quotes

L516: system thinking -> system**s** thinking

L519: alternatives imagined by human imagination. -> alternatives **through** human imagination.

L528-531: The use of quotation marks look a bit messy here. Please use single quotes within double quotes.

consists in the

L534: consists in -> consists **of**

L534-35:    "...contextualization of resilience research and discourse, that is, in embedding **it** in **its** political and cultural context" => as there are two different aspects the authors are referring two - research ad discourse - shouldnt it be "embedding **them** in **their** political and..."

L538 **a** change / change**s ;** system**s** thinking

L539-41: The sentence is confusing: Does this make more sense: For instance, on the one hand, environmental  movements  include stakeholders who develop a leverage required to transform established systems (such as energy systems) and their governance arrangements.

L542: hold power**,** thanks to such arrangements**,** typically

L551: ensure**s**

L557: tipping point for igniting transformation: There may be a more appropriate way to express this - as tipping points refer to a "situation, process or system beyond which a signiicant and unstoppable effect or change takes place" - and hence is more in line with describing an undesirable situation. Consider revising the phrase, perhaps, "and the threshold that needs to be surpassed for adaptation to to be considered as transformational."

L571: "conflict organized around scarce resources"? conflicts can happen without being intentionally organized. I would request the authors to consider using a different word, perhaps *conflict surrounding / related to ,,,*

L572: closing -> shutting down

L573: " becomes a more urgent research topic" -> become more urgent research topics

L575... "In other words, what are the implications of the disintegration of old systems for societies, that is, for their cultures, collective identities, traditions, economies, political-administrative power constellations, class structures, etc.?; and which societal transformations promote such disintegration?. In other words, what are the implications of the disintegration"->

"**W**hat are the implications of the disintegration of old systems for societies, that is, for their cultures, collective identities, traditions, economies, political-administrative power constellations, class structures, etc.? **W**hich societal transformations promote such disintegration?

L605: (unhabitat.org, 2019) -> please check if this is the correct way of citing internet sources for this journal.

L608: whom? **W**hat... { for all other instances please remove the semi-colons and start a new question.

L610-12: *excluded* -> **are being** excluded , comma after 'process' , involve -> involve**s**

  L621: A long debate has been going on for the past 2 decades about terminologies such as climate-induced migration, and the like, because of the oversimplification of complex phenomena, and problems with solely attributing an event of movement to climate change impacts. That is not to say that people do not move in response to environmental and/or impacts of climate change. However, at least the authors should consider using the term "mobilities" instead of migration, to include more than just discrete events of migration.

See the comment by:

Boas, I., Farbotko, C., Adams, H., Sterly, H., Bush, S., van der Geest, K., ... & Hulme, M. (2019). Climate migration myths. Nature Climate Change, 9(12), 901-903.

L621: semi-colon after the question mark should be deleted.

626: Very interesting point made here. However, do the authors actually mean ethnic relations rather than "racial relations".?

640: some decade**s** ago

649-650: the challenges of climate change-> the challenges of **addressing** climate change **impacts**

L653: system**s** thinking

L658: destroying -> **eliminating**

L661: consists in -> consists **of**

L663: The interaction  as well as the blurry line

In the conclusion, a final conclusive sentence is missing. I would suggest that the authors include one final sentence at the end of the conclusion.

---

## Author Response (AR2)

Reply to referee 3

Thank you for your careful reading of the text and we are truly sorry that it made for tedious reading. We have thoroughly re-written the text, bringing in the much needed nuances and getting rid of the irritating generalizations. The descriptive style could not be completely removed given the nature of the article. However, an argumentative style has also been included, unsubstantiated claims have been either removed or backed up by sources. The tendency towards dualism or polarization has also been corrected. We have also included a sample of the body of scholarship that mediates between 'naturalism' and 'constructivism' (in their strict or extreme senses). We have accepted most of the changes suggested by referee 3. The result is a trimmed text, without too much repetition.

As noted above, the whole text has been rewritten. More specifically, the whole introduction has been rewritten in such a manner that it is more argumentative and that the impression of polemic or dichotomy is avoided. We now speak of a dialectical field between naturalism and constructivism. Sections 2.1 (the 'naturalist' view) and 2.2 (the 'constructivist' view) have been merged and rewritten into a new whole. This means that there are many approaches and cross-fertilizations in that field, though there is still a 'naturalist' (in the strict sense) scholarship in the social sciences, that is to say, scholarship that draws on the methodology and conceptual framework common in (applied) natural sciences. We avoid speaking in terms of the 'naturalist' social scientist or the 'constructivist' social scientist.

**In line 307 you may be interested in earlier origins, as Indirli 2019 reviews them: historical flight and some open questions towards a pluralistic but holistic view of resilience**

Thank you for the additional reference. After line 307, we do mention other origins.

**Lines 455: "Naturalist social studies are based on the cybernetic idea"**

**Which are those "Naturalist social studies"? How did you methodologically assess it? And how do you know which variants exist? Why should they only follow a cybernetic approach??**

These sentences have been removed.

**Passages such as lines 758-762 are fine in general diction, but it is again not supported by sources, and the sources and claims in the sources following do not directly support this.**

This passage has been rewritten.

**Check wording in line 769, "into adaptive resilience tens to leave"**

Thanks!                                                                                                  Typo.

**Lines 849ff: How did you decide and justify to select smart urbanism and AI among many technological                                                                                                 trends?**
**Why is the whole section than not about "smart urbanism" at all?**

That section has been removed.

**Provide sources for your claims in lines 851-852. And again; "all" have drafted their AI strategies so? A 190 countries you have checked?**

**The presentation of AI is one-sided and misses all the critique from natural sciences on the failures of AI, the software crisis in the 1980s related to it already, etc. Again, this is rather an iteration of black and white stereotyping, missing structure, counterarguments and balance.**

Has been removed.

**Claims such as in lines 859-860 are again, worrisome; why should "Strengthening adaptive resilience to climate change through AI primarily" really only have resulted in "means that an integrated data system for circulating information (near) real time among agents needs to be developed" AI can do much more than just "circulating information"?!**

**That AI section still does not fit into the article. Maybe consider keeping it for a future article? Or embed it with more argumentation, as I tried to indicate.**

We have removed that section.

**Lines 994 following. The first sentence seems to be contradicted by the following sentence. And it would be good to see literature cited here that actually already tried to provide such mediation.**

Thanks. It has been rewritten.

**In section 4 it is a bit difficult to understand and find the six themes exactly. The beginning s of the sentences to each new theme could be made simpler to understand what are the main aspects of it. An example is in line 1016; it comes with.... but what do you mean? The whole section is rather long and once more, descriptive, narrative, with claims mixed with source-based review. The whole article is too much of the same in style. In 4, shorter would be better. Also, at least the first three themes, but actually, all of them simply repeat the previous sections.**

The whole section has been rewritten and trimmed. However, there is bound to be some kind of continuity with rest of the article since we believe that particular ongoing or new lines of research should still be pursued. But we also make it clearer why they should be.

**What is this dangling text from lines 1220 onwards? Content-wise it is really interesting and a bit novel and would warrant a paper. But as it is it is unclear what this part of text does.**

That text has also been removed.

**Line 1252 please avoid such jargon as "good old notion"**

Done.

**Only towards the end, the article comes back to climate change. Given the job background of most authors, it seems that this is the real intention, stated in line 1281 "are all connected to the issue of the political-administrative response" Maybe, the paper could be rewritten from the perspective of such governance; how it deals with resilience and these six fields? Or at least, be more explicit on why administrations increasingly become interested in such questions? This would be of benefit to scientific discussion. But it is also ok I think, if you keep it and only make it more balanced and avoid one-sided claims.**

We have tried to avoid one-sided claims. This suggestion is interesting, but for now not the right approach given the different backgrounds of the authors. But we shall bear it in mind for a future article.

**Towards the end of the conclusion, the article meanders into speculations about giant tech companies. It is alright to use a conclusion to expand the horizon of the article, but here as at other parts, it sounds a bit much like a journalistic speculative style.**

Has been removed/rewritten.

**In the rebuttal letter, the authors indicate to uptake some of the suggestions. I am not sure they**

**have met the expectations of the first reviewers. One recommendation I would like to reiterate and this time request the authors to deliver it (maybe I just did not find them): the text could well be cut on many text parts and benefit greatly from tables and maybe, framework charts that summarise the findings, find criteria for them and therefore, provide something novel to readers.**

**One final request; I know I generate a lot more work for you, my apologies. I generally like the overview this paper provides, it is helpful as a guidance for readers unfamiliar with the discussions. But especially for them, it is of great importance not to be guided in an unbalanced way**.

The general aim of the rewriting has been to bring in that needed balance. So thank you for your critical comments. We trust that we have achieved that goal.

[revised manuscript text omitted]

---

## Author Response (AR3)

Reply to reviewer:

Thank you so much for your meticulous reading of the revised manuscript and valuable suggestions, which we have incorporated in the text. There are two small exceptions and we would like to mention them briefly. One is the phrase 'consist in', which we have not changed to 'consist of'. In the first case, the notion of 'the essence of … is…' is being conveyed (see https://dictionary.cambridge.org/dictionary/english/consist-in-sth).

The second exception is L 462: it is here about proposing something (and hence no 'that').